# Phytochemical Evaluation of Tinctures and Essential Oil Obtained from *Satureja montana* Herb

**DOI:** 10.3390/molecules25204763

**Published:** 2020-10-16

**Authors:** Nataliia Hudz, Ewa Makowicz, Mariia Shanaida, Marietta Białoń, Izabela Jasicka-Misiak, Oksana Yezerska, Liudmyla Svydenko, Piotr Paweł Wieczorek

**Affiliations:** 1Department of Drug Technology and Biopharmacy, Danylo Halytsky Lviv National Medical University, 79010 Lviv, Ukraine; o.yezerska@gmail.com; 2Department of Analytical and Ecological Chemistry, University of Opole, 45-052 Opole, Poland; emakowicz@uni.opole.pl (E.M.); marietta.bialon@uni.opole.pl (M.B.); izabela.jasicka@uni.opole.pl (I.J.-M.); pwiecz@uni.opole.pl (P.P.W.); 3Department of Pharmacognosy and Medical Botany, I. Horbachevsky Ternopil National Medical University, 46001 Ternopil, Ukraine; shanayda@tdmu.edu.ua; 4Sector of Mobilization and Conservation of Plant Resources of the Rice Institute of the NAAS, Plodove, Kherson region, 74992 Kherson, Ukraine; svid65@ukr.net

**Keywords:** *Satureja montana*, tincture, polyphenols, flavonoids, essential oil

## Abstract

Winter Savory (*Satureja montana* L.) has been used in traditional medicine and as a spice or natural food preservative in the Mediterranean region for centuries. In this paper, some technological and analytical aspects of the *S. montana* tinctures development and an evaluation of the essential oil composition are provided. The total phenolic and flavonoid contents and phenolic compounds profile analyzed spectrophotometrically and by high-performance thin-layer chromatography (HPTLC), respectively, were evaluated in the developed tinctures. The results showed that the tinctures prepared from the *S. montana* herb by maceration or remaceration are rich in polyphenols, and there is an influence of the technological factors (particle size and extraction mode) on the total phenolic and flavonoid contents. Caffeic, rosmarinic, and chlorogenic acids, (–)-catechin and rutin were identified in the tinctures using the HPTLC method. *p*-Thymol (81.79%) revealed by gas chromatography-mass spectrometry (GC-MS) was the predominant compound of the essential oil of this plant. Thus, the high contents of polyphenols and flavonoids in the developed tinctures and *p*-thymol among the volatile components of the *S. montana* essential oil could indicate the promising antioxidant and antimicrobial properties of these herbal preparations. The obtained results are a ground for the organization of the manufacture of the *S. montana* tincture and essential oil with the purpose of performing preclinical studies.

## 1. Introduction

Winter Savory or Mountain Savory (*Satureja montana* L.) is a perennial semi-shrub that inhabits arid, sunny, and rocky regions. It belongs to the *Lamiaceae* family, *Nepetoideae* subfamily, *Mentheae* tribe [1,2,3,4,5]. *S. montana* is native to the Mediterranean area, but it is cultivated throughout the Europe [3,6,7]. Due to a specific pleasant odour, many *Satureja* species are used in food, cosmetic and pharmaceutical industries [2,5]. Among 13 observed species of the *Satureja* genus it was revealed that *S. montana* is one of the most pharmacologically active [8]. *S. montana* is used in Mediterranean cuisine [3] and traditional medicine for the treatment of respiratory system inflammation [9].

Extracts and essential oil from *S. montana* herb possess antibacterial, antiviral, antioxidant, anticatarrhal, antitumor, stimulant, expectorant, and other pharmacological activities [4,5,6,8,10]. Essential oils of many *Satureja* species contain oxygenates terpenes. Among them are thymol and carvacrol, which are the indicators of antimicrobial activity [5,6]. Due to such biological activities, *S. montana* could be regarded as a promising source for the manufacture of food products with antioxidant effects and herbal preparations with antimicrobial and antioxidant activities [5].

Studies of the antioxidant activity of different *S. montana* extracts (volatile and nonvolatile fractions) obtained by supercritical extraction and by hydrodistillation and soxhlet extraction allowed to compare the most known methods of obtaining volatile and essential oils, and DPPH and Rancimat methods. The volatile oil obtained by supercritical extraction was more active in scavenging the DPPH radical or inhibiting the lipid oxidation as it contained thymoquinone in 15 times higher than the essential oil obtained by hydrodistillation. Thymoquinone has antioxidant, neuroprotective, and anti-cancer activities [11].

Moreover, the resistance of microorganisms to known antimicrobial medicines provokes a search for such products among herbal products. Additionally, *S. montana* herbal preparations could be introduced into medicinal products as excipients with the purpose of the correction of their odour and taste. It is known that essential oils rich in carvacrol impart oregano odour, oils rich in *p*-cymene and thymol impart thyme odour, and oils rich in linalool impart lavender odour [5].

Phytochemical analysis of aerial parts of *S. montana* showed the presence of phenolic compounds, including flavonoids, and hydroxycinnamic acids, and compounds of essential oil (monoterpenes, sesquiterpenes, diterpens, etc.) [1,6,12]. Plant phenolics are a highly diversified group of compounds, which comprises simple phenolics, phenolic acids (hydroxycinnamic and hydroxybenzoic acids derivates), lignans, lignins, coumarins, flavonoids, stilbens, flavolignans, and tannins, phenolic terpenes, etc. [12,13]. Phenolics are very important plant antioxidants which inhibit undesirable oxidative processes in food products and the human body [12,14,15,16]. For that reason, the identification and quantification of phenolic compounds could be regarded as a tool related to potential of food and health benefits and accordingly as a quality index of elaborated tinctures [12,15].

According to the literature data, flavanone, flavone and flavonol glycosides are the main chemical groups identified in *Satureja* species [1,5,12]. Among the identified flavonoids of *S. montana* are luteolin-7-rhamnoside-4′-*O*-β-glucopyranoside [1], quercetin-3-*O*-α-l-rhamnopyranoside [1], quercetin-7-*O*-glucopyranoside [1], luteolin-7-*O*-glucopyranoside [1], 5-hydroxy-6,7,8,4′-tetramethoxyflavone [1], quercetin [1,9], catechin and epicatechin [12], luteolin [2], and rutin [9]. Among phenolic acids of *S. montana* were identified rosmarinic acid [2,5,9], caffeic acid [2,5,12], *p*-coumaric acid [12], ferulic acid [12], protocatechuic acid [5,12], syringic acid [9,12], vanillic acid [5,12], gallic acid [1], and ellagic acid [9].

Many studies demonstrated the antioxidant activity of *S. montana* extracts prepared with using different solvents [2,4,14] and essential oil [2,4,10,17]. *S. montana* is rich in essential oil, and its composition depends on many factors such as genetic features, geographical conditions, cultivation, and the development stage at harvesting time [2,18,19].

It was stated that essential oil and methanol or ethanol extracts of *S. montana* show significant antimicrobial activity of different levels against *Staphylococcus aureus* [1,10,19], *Streptococcus pyogenes* [10], *Escherichia coli* [1,4,10,19], *Salmonella typhimurium* [4,19], *Pseudomonas aeruginosa* [1,10,19], *Pseudomonas putida* [4], *Salmonella enteridis* [19], *Klebsiella pneumoniae* [10], *Aspergillus fumigatus* [10], *Aspergillus restrictus* [10], and other strains [4,5,10,19]. *S. montana* shows antiviral activity against several types of viruses, including human immunodeficiency virus type-1 (HIV-1) and *Cucumber mosaic* virus (CMV) [5]. Essential oil of *S. montana* significantly decreased the aflatoxin production by *Aspergillus flavus* [20]. Additionally, *S. montana* aqueous extracts did not show antibacterial activity. Considering antimicrobial performance of the essential oil and ethanol extracts of *Satureja montana*, they could be regarded as components of medicinal products in the dosage forms like oral liquids and drops for the treatment of foodborne diseases, and sprays for the treatment of wounds, infectious diseases of the oral cavity and other infections.

Herbal preparations are very attractive in the food industry as spices and additives in the modern phytotherapy [5,10,12,14,18]. The European Pharmacopeia distinguished three types of extracts from herbal substances according to their consistency: liquid (liquid extracts and tinctures), semi-solid (soft extracts and oleoresins), and solid (dry extracts). Tinctures are prepared by maceration or percolation using only ethanol of a suitable concentration for extraction of the herbal drug. They are prepared without use of heating that allows to save unstable and volatile compounds [21].

To the best of our knowledge, there are no reports on the development and standardization of *S. montana* tinctures as herbal medicinal preparations perspective for the treatment of infectious diseases of intestinal and respiratory systems, and food products for a general health improvement. Therefore, the primary purpose of our study was to develop the *S. montana* tinctures and evaluate their total phenolic and flavonoid contents depending on different technological factors, including particle size, extraction mode, and year of herb collection as well. The secondary purpose was to evaluate chromatographic fingerprints of the tinctures by HPTLC method and conduct GC-MS analysis of the essential oil of *S. montana*. The identification of phenolics can provide useful information relating to antioxidant and health benefits of herbal medicines.

## 2. Results and Discussion

Ultraviolet–visible spectrophotometric methods are widely used in pharmaceutical analysis. These methods are low-cost, easy, rapid, and applicable in routine control of herbal substances and herbal preparations. The Folin-Ciocalteu method is pharmacopeia one used for the determination of tannins. Aluminum chloride colorimetric method is widely used for the determination of TFC [21].

TPC and TFC could be considered important indices for evaluating quality of the tinctures. TPC and TFC were measured in the developed tinctures of *S. montana* herb spectrophotometrically. The calibration curves of the two reference standards for the determination of TPC are provided on Figure 1.

According to the conception of analytical quality by design, time is considered as one of the critical method parameters in the analytical procedure of the determination of total phenolics by means of the spectrophotometric method with Folin–Ciocalteu reagent (FCR) as the reaction time has an impact on the performance of this method [22,23,24,25]. Moreover, changes in time can be regarded also as a deliberate variation in the method parameters. Such a variation is also related to the robustness of the analytical procedure and should be evaluated in the development of this analytical procedure [26,27]. Therefore, we studied the influence of the reaction time for the two tinctures with FCR on the TPC.

The results of the kinetics reaction study of FCR with the *S. montana* polyphenols show that there is a high (0.9 > r > 0.7)) or very high correlation (r > 0.9) between the absorbance and time up to 62–70 min (Table 1).

After incubation for 62–70 min the absorbance increases insignificantly, namely a time of 66 ± 4 min is enough for the complete reaction of polyphenols with the FCR. Further measurements showed that increasing in a reaction time did not induce a significant increase in the absorbance. The square correlation coefficient (R^2^) of the absorbance dependence on the reaction time for 70–92 min was in the range of 0.8214 to 0.9686. The measurements at a wavelength of 760 nm for more 92 min give its insignificant elevation and a significant decrease of R^2^, for example, it was 0.7745 for 118 min, 0.6347 for 147 min for tincture T2. In fact, there is an insignificant increase in the absorbance (approximately +1%) for 92 min compared with the absorbance for 65 min for tincture T2. Similar effect was observed for tincture T1. According to the requirements of the State Pharmacopeia of Ukraine [27], such a deviation is considered insignificant at the limits of active substances in the range of 90% to 110% of the stated content. Two experiments for tincture T1 confirmed the repeatability of the results.

Such regularity was established in the performed studies: the longer was the reaction time, the less was the R^2^ between the absorbance and time of the reaction of FCR with the tinctures of *S. montana.* In addition, 62 and 92 min can be chosen as the period for checking robustness of the analytical procedure of the TPC determination in the *S. montana* tinctures. Therefore, on the basis of experimental studies, it was set up and justified that 65 min is the time of the almost complete interaction of *S. montana* polyphenols with the FCR and reaction time more than 65 min has no influence on the TPC determination. The TPC is provided in Table 2.

Table 2 shows the TPC expressed as per 1 L of a tincture and per g of dry weight of the herbal substance depending on the particle size. The obtained results confirmed the influence of the particle size on the TPC: the less the particle size, the higher the TPC in the tinctures or by other words: the smaller the particle size, the more effective is the phenolics extraction. In our studies the TPC was 22.7 mg and 13.0 mg of gallic acid equivalents per 1 g of the dry herbal substance, and 45.63 and 26.16 mg of rutin equivalents per g of the dry herbal substance, respectively, for the tinctures obtained from the herbal substance crushed to 1–3 mm and 3–5 mm, respectively. Hajdari et al. [6] stated that the concentration of TPC ranged from 68.1 to 102.6 mg caffeic acid equivalent/g of plant dry mass. Ćetkovic et al. [12] showed that the TPC was 8.36 µg, 969.43 µg, 1358.14 µg, and 96.36 µg of chlorogenic acid equivalent/g of plant dry mass if chloroform, ethyl acetate, *n*-butanol and water were used as solvents, respectively, for the extraction of phenolics from the methanol extract. Moreover, Ćetkovic G.S. et al. [12] showed that *n*-butanol extracted phenolics in the most amounts and chloroform did in the least ones. Chrpová et al. showed the TPC was 27.1 mg gallic acid equivalents per 1 g of plant dry mass if water was used as solvent at a temperature of 70 °C for 10 min. Hassanein et al. determined rosmarinic and caffeic acids contents in the ethanol extract: 1.0 and 1.7 mg per 1 g of dry herb of *S. montana* [2]. TPC in the 70% methanolic extract of S. *montana subsp. kitaibelii* herb measured by the Folin–Ciocalteu method was 25.82 mg gallic acid equivalents per 1 g of dry sample [28]. Total phenolic and flavonoid contents in the *S. montana* herb was higher in methanol extract compared to the chloroform one [29]. Therefore, it is almost impossible to compare these results as the different reference standards, solvents, particle size, ratio of drug to herbal preparation, and extraction techniques were used despite the usage of the same method for the TPC determination. It seems that the results obtained in our study are close to the results provided by López-Cobo et al. [28].

López-Cobo et al. and Jakovljević et al. showed that water, ethanol in different concentrations 30%, 50%, 70%, and methanol extracted phenolic acids and flavonoids in the form of aglycons and glycosides. Moreover, the extent of the extraction depends on a solvent, its concentration, temperature, nature of a phenolic compound [28,30]. Furthermore, different authors differ in their opinions on an optimal solvent. However, only ethanol of various concentrations usually at room temperature is used in the pharmaceutical industry for the tincture manufacture [21,27].

The determination of the TFC was used for the quality evaluation of the elaborated tinctures as flavonoids are known as plant secondary metabolites with known antioxidant activity [31]. The TFC was determined by the aluminum chloride colorimetric method which is described by Chan et al. [32] and Hudz et al. [33]. The linearity of the elaborated procedure was proved in the rutin concentration range of 10–100 mg/L (Figure 2).

The wavelengths of maximum absorption of rutin and the developed tinctures were used for the measurements of the absorbance for the TFC calculation. It was revealed that rutin trihydrate and tinctures of *S. montana* herb have the maximum absorption at a wavelength of 411 ± 1 nm and in the range of 386–390 nm, respectively, in their differential spectra (Figure 3).

A hypsochromic shift of the tinctures compared to rutin could be explained by the presence of flavones in *S. montana,* especially luteolin, apigenin, and their glycosides, having an absorption maximum in the range of 385–395 nm [2,5,28,32,33].

In further measurements instead of plotting a calibration curve, a point of 50 µL of a stock solution of rutin trihydrate (1000 mg/L) was used at least one time per day as a middle point from the linearity range. Moreover, according to the Guideline on specifications: test procedures and acceptance criteria for herbal substances, herbal preparations and herbal medicinal products/traditional herbal medicinal products, an overall control strategy includes in-process testing, stability testing and testing for consistency of batches [34]. Therefore, our study was directed at establishing the repeatability results of the TFC determination as well. The results of the TFC determination were expressed as mg rutin equivalent per 1 L of a tincture and 1 g of the herbal substance and presented in Table 3.

As can be seen from Table 3, the year of the herb collection does not have significant influence on the TFC at a particle size of 3–5 mm while the particle size of *S. montana* herb and extraction mode significantly influence the TFC in the tinctures. It was established that a noticeable decrease in the TFC observed in 14 days that needs further studies in order to set up a period during which chemical processes in tinctures of the herb *S. montana* stop after maceration and the herbal product is ready for releasing from a pharmaceutical factory. Null hypothesis testing was used for the evaluation of differences between two mean values of the TFC per 1 g of the herbal substance obtained for the tinctures which are differed in particle size of the herbal substance, extraction mode, data of the TFC determination, etc. In all the cases the null hypothesis (*H_0_*) was tested that the difference is equal to zero, namely two means were assumed to be the same at *H_0_* (*μ_1_* = *μ_2_*). Firstly, the statistical analysis revealed that the TFC in tinctures with a particle size of 1–3 mm is statistically higher in comparison to the ones with a particle size of 3–5 mm (lines 1, 3, and 6 of Table 4). Secondly, there was no effect of the storage time (7 months) of the herbal substance on the extraction degree of flavonoids (tinctures 7 and 3, line 2 in Table 4). Additionally, the year of the herb collection does not have considerable influence on the TFC at a particle size of 3–5 mm (lines 4 and 5 of Table 4). Finally, the statistical analysis proved significant difference of the TFC in tincture T5 determined after maceration immediately and in the next 14 days (line 9 of Table 4).

The influence of extraction mode could be explained by the following factors. More effective remaceration in time is not effective in extraction as maceration for 1 day yields a significantly small amount of extracted flavonoids (Table 5). As can be seen from Table 5, the more days of extraction, the higher is the TFC. Using maceration, 5–14 days of extraction is a general time of tinctures manufacture [35].

Thin-layer chromatography (TLC) and HPLC chromatograms could be considered as marker fingerprints that can identify a species and distinguish related species of plants [13]. Moreover, according to the Guideline on specifications: test procedures and acceptance criteria for herbal substances, herbal preparations and herbal medicinal products/traditional herbal medicinal products, if the use of a non-specific assay is justified, other supporting analytical procedures should be used to achieve overall specificity, namely if a UV/VIS spectrophotometric assay is used, a combination of this assay and a suitable test for identification (e.g., fingerprint chromatography) can be used [34]. Additionally, TLC is especially useful for screening experiments. TLC has some privileges and among them are a presentation of results as a picture-like image; a lot of samples can be compared side by side; all samples are analyzed under the same conditions on the same plate, which is impossible to perform in case of a sequential mode of HPLC. Fast results are obtained as a lot of samples are analyzed at the same time that leads to solvent savings. However, one of the greatest drawbacks of TLC-based effect-directed analysis is its rather low separation capacity compared to HPLC. The observed bands can be a mixture of different compounds [36]. Therefore, it should be underlined that TLC analysis is only an additional tool which is useful to support non-specific assay, for example, an assay of flavonoids by a colorimetric method with AlCl_3_ and distinguish related species.

HPTLC separation was performed according to the analytical procedure developed by N. Stanek and I. Jasicka-Misiak [16]. The HPTLC fingerprint analysis allowed to identify phenolic compounds in the tincture of *S. montana* herb T5 obtained with maceration at a particle size of 1 to 3 mm (year of collection 2018).

The fingerprintis with their color description for tincture T5 before and after derivatisation are presented in Table 6 and Figure 4, Figure 5 and Figure 6.

The HPTLC fingerprints of this tincture had two prominent bands at R_f_ = 0.34 (light blue) and R_f_ = 0.52 (blue) under 254 nm and 366 nm before and after derivatisation that corresponded to rosmarinic and caffeic acids, respectively. The complex identification by means of colour reactions at 254 and 366 nm irradiation before and after derivatisation did not allow to unambiguously differ luteolin from quercetin in the tincture due to low separation. As illustrated in Figure 5, very similar Rf are observed for luteolin (R_f_ = 0.54) and quercetin (R_f_ = 0.56). Under 254 nm irradiation before and after derivatisation, respectively, there is a black band of low intensity and deep yellow band of low intensity (R_f_ ~ 0.55) that could meet Rf of the reference standard of luteolin and quercetin. Moreover, the chromatograms exhibit the two notable bands of intensive blue and intensive black colours at R_f_ = 0.27 − 0.28 and 0.25, respectively, under 254 nm and 366 nm irradiation. One of them (R_f_ = 0.25) could be ascribed to (–)-catechin. The band of the reference standard of (–)-catechin has the lowest intensity among all the three phenolic compounds taken in a concentration of 0.1 mg/mL. Additionally, in case of rutin (R_f_ = 0.03) and chlorogenic acid (R_f_ = 0.09), it seems that they are present in the tincture in a small amount. Furthermore, in case of apigenin (R_f_ = 0.58), it is supposed that it is not present in a detectable concentration in this tincture.

The obtained results are partly in line with the published studies earlier [2,9,30]. Kremer et al. [9] identified six phenolic compounds (rutin, quercetin, *p*-coumaric, ellagic, rosmarinic and syringic acids) using HPLC in methanolic and ethanolic extracts of *S. montana.* Amount of rosmarinic acid as a predominant compound of *S. montana* herb (0.21–7.84 µg/mg) depends on the used technological ways and solvents [30]. In ethylacetate and *n*-butanol extracts of *S. montana* the hydroxybenzoic acid derivatives (protocatechuic, syringic and vanilic acids), hydroxycinnamic acid derivatives (caffeic, *p*-coumaric, and ferulic acids), (±)-catechin and (–)-epicatechin were identified [12]. Rosmarinic acid, caffeic acid and luteolin were identified in the extracts obtained with the aid of 80% ethanol from *S. montana* collected in Egypt [2].

The obtained identified and unidentified bands on HPTLC plates create a unique pattern of the *S. montana* tincture. As we observed in our unpublished research, three species of the *Lamiaceae* family (*Satureja montana, Salvia officinalis,* and *Thymus vulgaris*) collected in the same location and analyzed in the same plate create their own unique fingerprints. However, one of the major drawbacks of this study is the evaluation of only one the *S. montana* tincture in the HPTLC method. However, there is an advantage as the analytical procedure is developed and interpreted. This procedure can be supplemented by other species of the *Lamiaceae* family, reference standards, and modified by their concentrations.

*S. montana* L. essential oil was subjected to a detailed analysis to determine its chemical composition. As shown in Table 7, 34 compounds were identified. According to the obtained analytical data, *S. montana* had an unusual chemical profile. *p*-Thymol as an isomer of thymol and carvacrol was dominated (about 81.79%) (Table 7, and Figure 7 and Figure 8). This compound has not been stated earlier in the available scientific publications as the dominant component of the *S. montana* essential oil. Thus, the existence of a special chemotype of *S. montana* grown on the experimental plots in the Kherson region was revealed. Among the other components revealed in concentrations more than 1% were linalool (2.09%), 1-octen-3-ol (1.91%), γ-terpinene (1.65%), *o*-cymene (1.26%), *cis-β*-terpineol (1.21%), and terpinen-4-ol (1.07%).

According to literature data, essential oils in which such aromatic substances as carvacrol, eugenol or thymol dominate have pronounced antimicrobial activity [4,5,37,38]. Hajdari et al. [6] stated absence of carvacrol in the myrcene and viridiflorol chemotype and linalool chemotype. These chemotypes contained also minor amounts of thymol (0.15–0.79%). Additionally, the chemotypes which contain one compound in a very high concentration were described by Nemati et al. (carvacrol 83.40%), Hassanein et al. (carvacrol 79.75%), Miladi et al. (carvacrol 53.35%), and Hajdari et al. (linalool 50.42%) [2,6,17,19].

Literature data showed the discrepancies in the chemical composition of essential oils of *S. montana*. According to the different published data, essential oil of *S. montana* consists mainly of carvacrol (up to 0.1–83.40%), *p*-cymene (0.66–41.4%), thymol (0.15–46%), linalool (0.1–50.42%), and other monoterpenoids, as well as sesquiterpenes, and diterpenes [1,2,4,6,10,39]. These differences are induced with the change in environmental impact (altitude and microclimatic conditions) on the plant [6,10], the phase of collection, the variability within the same species caused with the presence of different chemotypes [6], and country of location [2,6]. Essential oils of *S. montana* subsp. *variegata* and subsp. *montana* analyzed by GC-MS spectrometry showed the domination of carvacrol (22.5%), *p*-cymene (17.6%), and thymol (17,4%) in the first subspecies, and carvacrol (61.9%), *p*-cymene (9,9%) and γ-terpinene (8.2%) in the second subspecies [39]. For example, Mihajilov-Krstev et al. [10] stated that the content of the major aromatic phenol compounds (thymol and carvacrol) significantly decreased with the increase in the altitude of growing. It was 24.69% and 2.46% for an altitude of 100 m, 2.46% and 24.46% for an altitude of 500 m and 0% for an altitude of 800 m, respectively, for thymol and carvacrol. The aerial parts of wild growing *S. montana* ssp *montana* were collected during the flowering stage from the three localities in Montenegro differed in the altitude of their growing [10].

Similar results were stated by Hajdari et al. [6] for *S. montana* collected in Albania (Valbonë location) and Montenegro (Ulcinj location). Thymol (31.1%), carvacrol (16.2%) and *p*-cymene (13.7%) were found in the essential oil of *S. montana* collected in Valbonë. Thymol (14.97%), carvacrol (17.92%) and *p*-cymene (26.14%) were found in the essential oil *S. montana* collected in Ulcinj. Hajdari et al. [6] consider that plants from Valbonë and Ulcinj belong to the monoterpene aromatic polyphenol chemotype (*p*-cymene, thymol and carvacrol chemotype). Aerial parts of *S. montana* were collected from July to September 2013 in the locations from naturally growing populations in Albania and Montenegro [6]. Carvacrol and thymol are responsible for high antimicrobial activity of essential oils of *S. montana* and *Thymus vulgaris* [19,40]. According to B. Tepe and M. Cikiz, carvacrol is the main indicator of antimicrobial activity of essential oil [5].

Therefore, currently there are known some chemotypes, based on the chemical composition of the essential oil analysis of *S. montana* [6,10,20]. Among them are chemotype with thymol and carvacrol domination [4], carvacrol [1,2,17], chemotype of linalool, *p*-cymene and α-terpineol [4], chemotype of linalool [6,10], intermediate chemotype of *p*-cymene/linalool [6], chemotype of *p*-cymene, thymol and carvacrol [6], chemotype of *p*-cymene and carvacrol [19], chemotype of myrcene and viridiflorol [6], chemotype of linalool, thymol and carvacrol [10], chemotype of *p*-cymene, linalool, thymol and carvacrol [3], chemotype of carvacrol, γ-terpinene, and *p*-cymene [20].

## 3. Materials and Methods

### 3.1. Plant Material

Aerial part of *S. montana* was collected in the flowering stage in Kherson region (Ukraine) in 2017 and 2018. Voucher specimens of each year were deposited at the Herbarium of the Sector of Mobilization and Conservation of Plant Resources of the Rice Institute of the NAAS (Plodove, Kherson region, Ukraine) and at the Department of Analytical and Ecological Chemistry of University of Opole (Poland). The aerial part was dried and kept at room temperature in dark place before the preparation of the tinctures and essential oil.

### 3.2. Extraction

All the tinctures were obtained in a ratio of the herbal substance to a final product as approximately 1 to 10. As a solvent, 70% ethanol was used. The herbal substance was reduced to pieces. The crushed herbal substance was sieved through suitable sieves with the size of holes of 1, 2, 3, 5, and 7 mm. Then the ground herbal substance of the appropriate size was mixed with 70% ethanol. The mixtures stood in closed containers. Maceration was performed at room temperature for 7 days. After this period the residue was separated from the extraction solvent by means of filtration through a paper filter. Remaceration included maceration for 24 h with the following three macerations for a period of 2 h for each one. Filtration was performed after each maceration and the obtained liquids were combined. Therefore, the total maceration was for a period of 30 h (24 h + 3 macerations × 2 h = 30 h).

The characteristics of the prepared tinctures are presented in Table 8.

The analytical procedures of the total phenolic and flavonoid contents determination and HPTLC fingerprint analysis were used for the quality evaluation of the elaborated tinctures of *S. montana* herb.

### 3.3. Essential Oil Isolation Procedure

Essential oil from dry *S. montana* herb (collection of 2017) was extracted via hydrodistillation for 4 h with a Clevenger-type apparatus. The oil was kept at 4 °C in a sealed amber bottle before analysis by GC-MS.

### 3.4. Chemicals and Reagents

All the chemicals and solvents used for analyses were of analytical purity grade. Dichlorometan, methanol, ethanol, chloroform, ethyl acetate, formic acid and aluminum chloride hexahydrate were purchased from POCH S.A. (Gliwice, Poland). The FCR and standards (apigenin, luteolin, caffeic acid, chlorogenic acid, gallic acid, rosmarinic acid, (–)-catechin, rutin, and quercetin were purchased from Sigma-Aldrich (Poznań, Poland)

### 3.5. Determination of Total Phenolic Content (TPC)

The TPC of the tinctures was determined by the Folin–Ciocalteu method using gallic acid and rutin as the reference standards according to the analytical procedure described in the publications [25,41]. Gallic acid was chosen as it had been identified as a phenolic compound in *S. montana* [1]. Rutin was chosen as commercially available reference standard. Moreover, rutin [9,30] and flavonoids with quercetin aglycon were identified in *S. montana* [28].

100 µL of dilution of a tincture (1:20 in 50% ethanol) were mixed with 100 µL of FCR, 1500 µL of purified water and 300 µL of 20% solution of sodium carbonate. The mixtures were mixed by vortex and incubation was done at room temperature at darkness for 2 h. Absorbance was measured at 760 mm with using spectrophotometer “Photometry Hitachi U-2810”. Purified water was used as a blank. The test was carried out for the two tinctures in triplicate. The test was performed twice for tincture T2 for the repeatability study of the elaborated analytical procedure. The results were expressed as gallic acid and rutin equivalents: mg eq-gallic acid and mg eq-rutin per 1 L of a tincture and per 1 g of the herbal substance. The curves of gallic acid and rutin hydrate were plotted in the concentration range of 20 to 150 mg/L and 62 to 310 mg/L, respectively. The stock solutions of gallic acid monohydrate (1100 mg/L) and rutin trihydrate (1200 mg/L) were prepared using purified water and 50% aqueous solution of ethanol, respectively. The TPC was calculated using expression C = c × 20 × k, where C is TPC of the tested tincture, c is TPC taken from the calibration curve, 20 is coefficient of dilution of the tincture for testing, k is coefficient for the calculation of rutin trihydrate into rutin (0.919) and 1 for gallic acid. The mean of three measurements was used for each concentration of the reference standard. The kinetics of the reaction for the tinctures 1 and 2 was evaluated by comparing r (√R^2^) between the absorbance at 760 nm and the reaction time. r ≥ 0.7 was established as the acceptance criterion [25].

### 3.6. Total Flavonoid Content (TFC)

TFC was determined using the slightly modified analytical procedures of differential spectrometry provided by A. Meda et al. for the estimation of the TFC in honey [42] and by N. Hudz et al. for bee bread [33]. Rutin trihydrate was used to build the calibration curve in the concentration range of 20 to 100 mg/L. Rutin trihydrate dissolution in 50% ethanol was carried out with the aid of ultrasound. The results were expressed as rutin equivalents: mg eq-rutin/L of a tincture and mg eq-rutin/g of the *S. montana* herb.

The TFC was calculated in two ways. The first way was used for tinctures T1 and T2 with the simultaneous study of linearity of the analytical procedure. Thereby, 10, 30, 50, 70, and 100 µL of the stock solution of rutin trihydrate (1000 mg/L) were diluted with 50% ethanol up to 1.0 mL. The obtained dilutions of rutin trihydrate were mixed with 1.0 mL of 2% aluminum chloride hexahydrate in 50% ethanol. After incubation at room temperature for 75 ± 10 min the spectra of the reaction mixtures were measured in the range of 360 nm to 460 nm with spectrophotometer (Photometry Hitachi U-2810). The volume of 2% aluminum chloride hexahydrate in 50% ethanol was substituted by the same volume of 50% ethanol in the blank for each dilution of rutin trihydrate. In a like manner, 50 µL of the developed tinctures of *S. montana* were diluted with 50% ethanol up to 1.0 mL and was mixed with 1.0 of 2% solution of aluminum chloride hexahydrate. The mixture was mixed by vortex and incubation was done at room temperature for 75 ± 10 min. The volume of 2% solution of aluminum chloride was substituted by the same amount of 50% ethanol in blank. The test was carried out for each tincture in triplicate. For tinctures T1 and T2 the TFC was calculated using expression C = c × 20 × k, where C is TFC of the tested tincture, c is TPC taken from the calibration curve, 20 is coefficient of dilution of the tincture for testing, k is coefficient for the recalculation of rutin trihydrate into rutin (0.917).

The second way was used for the other tinctures. Instead of plotting a calibration curve, a point of 50 µL of a stock solution of rutin trihydrate (1000 mg/L) was used at least one time per day. In this case TFC was calculated using the following formula: C = A*_test_* × *m* × 1000 × *k*/A*_rutin_* × V*_rutin_*, where C is TFC of the tested tincture (mg/L), A*_test_* is the absorbance of a reaction mixture of a tincture at the absorption maximum, A*_rutin_* is the absorbance of a reaction mixture of rutin solution in a concentration of 50 mg/L at the absorption maximum, m (mg) is mass of rutin trihydrate for the preparation of the stock solution (approximately 1000 mg/L), k is coefficient for the recalculation of rutin trihydrate into rutin (0.917), and V*_rutin_* is a volume in which rutin was dissolved.

### 3.7. HPTLC Analysis

HPTLC analysis of phenolic compounds was carried out using the CAMAG analytical system (Muttenz, Switzerland) as described by N. Stanek and I. Jasicka-Misiak [16]. The reference standards of phenolic compounds were dissolved in methanol in a concentration of 0.5 mg/mL with the exception of rutin, quercetin, and (–)-catechin. For these three reference standards, the concentration was 0.1 mg/mL. The tincture and standard solutions in a volume of 10 μL were applied to chromatographic plates (20 cm × 10 cm) using an automatic application device. The mobile phase consisting of chloroform: ethyl acetate: formic acid (5:4:1, *v*/*v*/*v*) was used for chromatography. Visualization was performed in two ways: (1) under UV light (254 nm, 366 nm) and (2) under UV light (254 nm, 366 nm) after spraying with 1% methanolic AlCl_3_ and drying for approximately 3 min in warm air. The obtained chromatographic results were analyzed using HPTLC software (vision CATS, CAMAG, Muttenz, Switzerland).

### 3.8. GC-MS Analysis of Essential Oil

The volatile compounds of the *S. montana* L. herb were identified by comparing the mass spectra data with spectrometer database of the NIST 11 Library and by comparison of their retention index calculated against *n*-alkanes (C_9_–C_20_). Each chromatographic analysis was repeated three times. The average value of the relative composition of the essential oil percentage was calculated from the peak areas. The Hewlett Packard HP 6890 series GC system chromatograph (Hewlett Packard, WALDBRONN, Germany) was used for the study, which was coupled with the Hewlett Packard 5973 mass selective detector (Hewlett Packard, Waldbronn, Germany). The chromatograph was equipped with the non-polar, high-temperature ZB-5HT (5% diphenyl- and 95% dimethylpolysiloxane) capillary column of length of 30 m, inner diameter of 0.32 mm, film thickness of 0.25 μm (Phenomenex Inc., Torrance, CA, USA). The gas chromatograph was equipped with a split injector; the split ratio was 20:1 and 1 μm of a sample was introduced. Helium served as the carrier gas, and its flow rate was 2 mL/min. Analyses were performed at the temperature range of 40–280 °C and the heating rate was 10 °C/min. Injector temperature was 250 °C.

### 3.9. Statistical Analysis

All analyses for each tincture were carried out in triplicate and the results were expressed as mean value ± standard deviation (SD) or mean value ± relative standard deviation (RSD). Correlation coefficients (r) to determine relationships between values were calculated using MS Excel Software. Achim Buyul and Peter Tsefel’s classification was used to evaluate correlation coefficients (r) between two magnitudes: up to 0.2 is very weak, up to 0.5 is weak, up to 0.7 is medium, up to 0.9 is high and over 0.9 is a very high correlation [43].

Wayne’s statistical analysis was employed for comparison of the TFC mean values determined in different days, for different particle sizes, etc. Decision rule in all the cases was: with α = 0.025 critical values of t* should be in the range of −2.78 to +2.78. *H*_0_ was rejected if *t** < −2.78 or *t** > +2.78 [33,44].

## 4. Conclusions

As a result of the performed studies, the tinctures from the *S. montana* herb were developed by different methods of maceration. It was revealed that the determination of the total phenolic and flavonoid contents could be used for the quality evaluation of tinctures from the *S. montana* herb. The tincture obtained by maceration possessed the highest content of phenolics compared to remaceration. It can be concluded that there is an influence of the particle size (1–3 mm versus 3–5 mm) and extraction mode (maceration and remaceration) on TPC and TFC in the tinctures while the herb storage for 18 months does not influence the TFC in the tinctures of the S. *montana* herb. Rosmarinic, caffeic and chlorogenic acids, rutin and (–)-catechin were identified in the tinctures of *S. montana* using the HPTLC screening method as an additional method to non-specific assay and for distinguishing from related species. The issue of unambiguous identification of luteolin and quercetin in the tincture is controversial because of their low separation by the elaborated HPLTC procedure. Generally, 34 compounds were identified in the essential oil of this plant by GC-MS, and *p*-thymol was dominated (81.79%). The high content of polyphenols in the tinctures and *p*-thymol in the essential oil could point the prominent antioxidant properties and antimicrobial potential of the obtained herbal preparations. Further studies are needed to evaluate the stability of the elaborated tinctures, especially in the first two weeks, and fingerprints of tinctures prepared from *S. montana* herb collected in different time periods, and establish antimicrobial activity and phenolic profile by high performance liquid chromatography. In general, the results of our study can be a basis for the development and standardization of the tinctures and essential oil of *S. montana* herb in industrial conditions. The essential oil and tinctures of *S. montana* could be used as components for preparation of oral liquids, drops and antimicrobial sprays for the treatment of foodborne diseases, infections of the oral cavity and respiratory tract, and wounds.

## Figures and Tables

**Figure 1 molecules-25-04763-f001:**
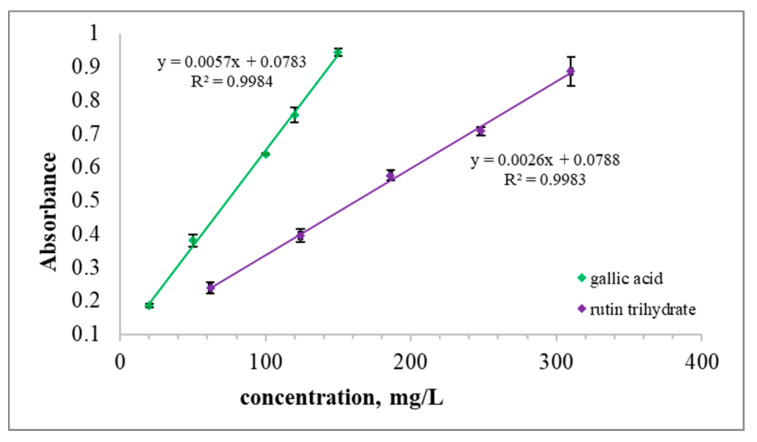
Calibration curves of rutin trihydrate and gallic acid for the determination of TPC in the *S. montana* tinctures.

**Figure 2 molecules-25-04763-f002:**
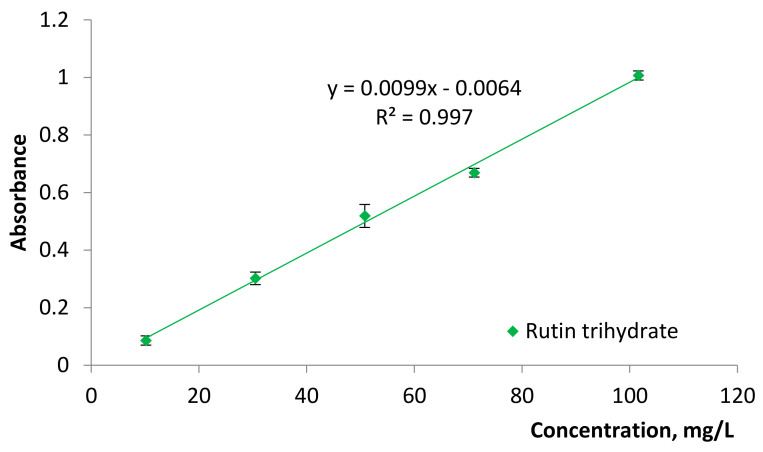
Calibration curve of rutin trihydrate for the determination of TFC.

**Figure 3 molecules-25-04763-f003:**
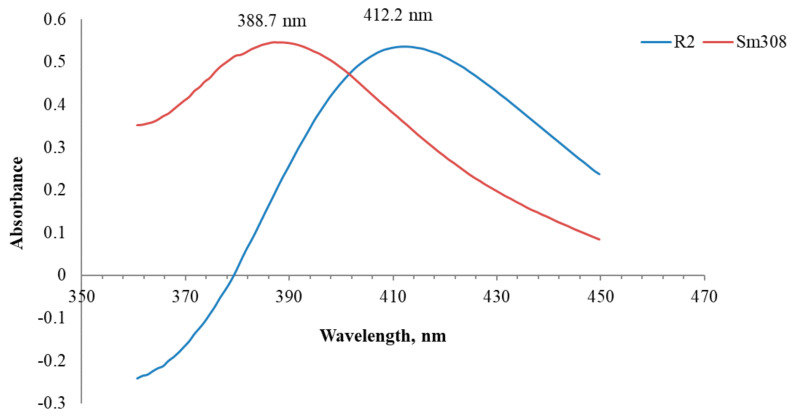
Differential spectra of rutin (R2) (50 mg/L) and *Satureja montana* tincture T5 (Sm308).

**Figure 4 molecules-25-04763-f004:**
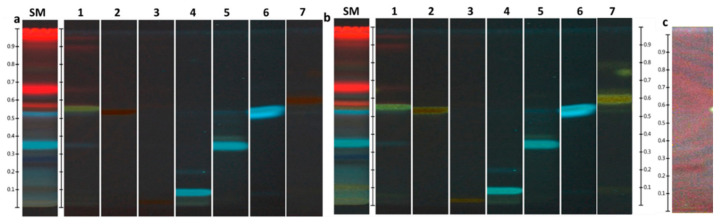
HPTLC fingerprints of *Satureja montana* tincture under 366 nm before (**a**) and after (**b**) derivatisation with 1% of AlCl_3_; SM—*S. montana* tincture; 1—quercetin; 2—luteolin; 3—rutin; 4—chlorogenic acid; 5—rosmarinic acid; 6—caffeic acid; 7—apigenin; (**c**) (–)-catechin.

**Figure 5 molecules-25-04763-f005:**
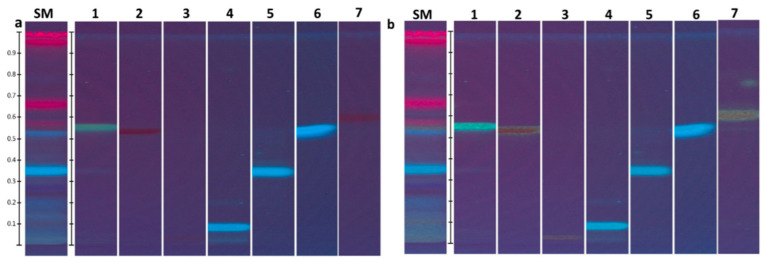
HPTLC fingerprints of *Satureja montana* tincture under 254 nm before (**a**) and after (**b**) derivatisation with 1% of AlCl_3_; SM—*S. montana* tincture; 1—quercetin; 2—luteolin; 3—rutin; 4—chlorogenic acid; 5—rosmarinic acid; 6—caffeic acid; 7—apigenin.

**Figure 6 molecules-25-04763-f006:**
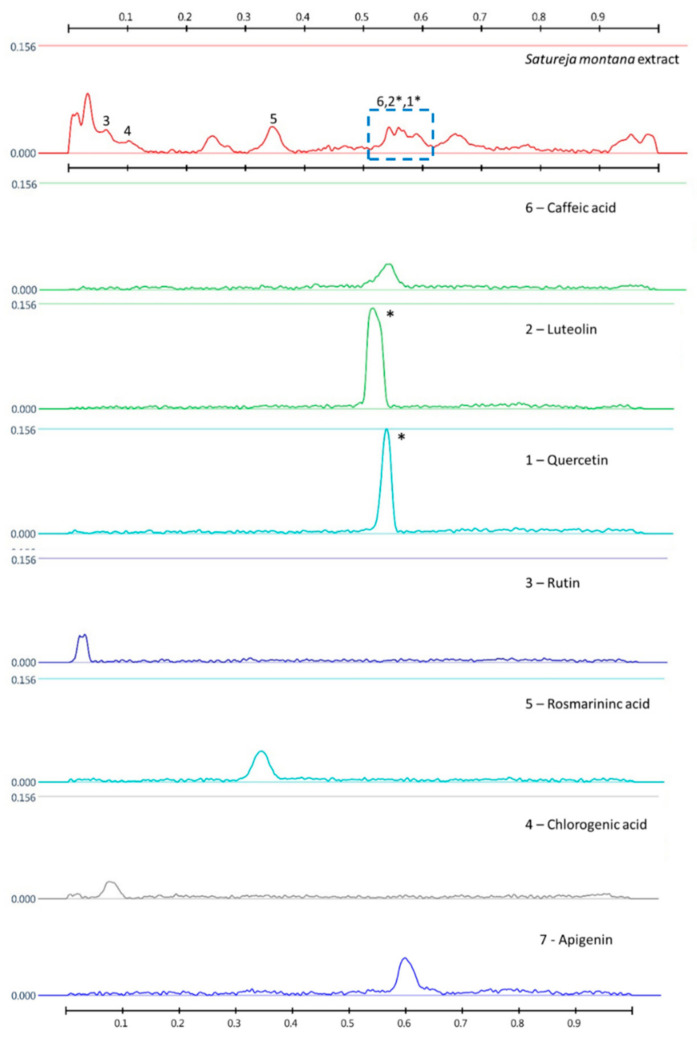
*Satureja montana* tincture and identified standards profiles on HPTLC chromatogram pictures at 366 nm after derivatisation; *—not possible unambiguous identification due to low separation.

**Figure 7 molecules-25-04763-f007:**
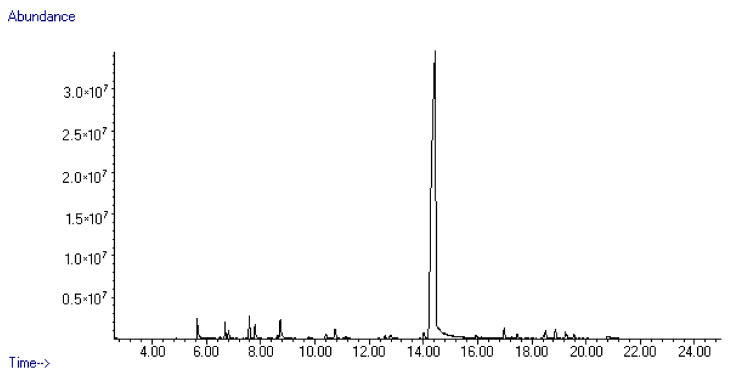
GC-MS chromatogram of the *Satureja montana* essential oil.

**Figure 8 molecules-25-04763-f008:**
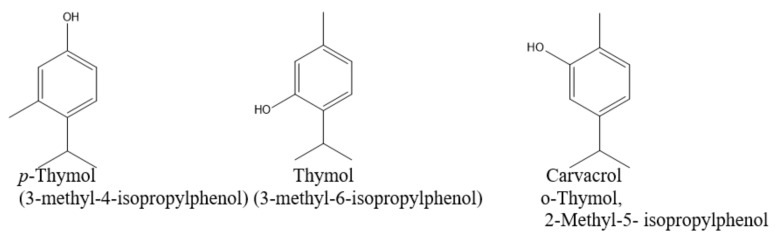
Chemical structures of the thymol isomers.

**Table 1 molecules-25-04763-t001:** Statistical data for the regression equation of the absorbance dependence on the reaction time in the TPC analysis of the *S. montana* tinctures.

T2			T1					
Time, min	Mean Absorbance ± SD	Correlation Equation, R^2^	Time, min	Mean Absorbance ± SD	Correlation Equation, R^2^	Time, min	Mean Absorbance ± SD	Correlation Equation, R^2^
30	0.426 ± 0.009	–	32	0.751 ± 0.039	–	34	0.781 ± 0.035	–
–	–	–	47	0.769 ± 0.032	y = 0.0012x + 0.7126 R^2^ = 1	–	–	–
65	0.436 ± 0.008	y = 0.0003x + 0.4174 R^2^ = 1	70	0.774 ± 0.025	y = 0.0006x + 0.7362 R^2^ = 0.8214	62	0.784 ± 0.028	y = 0.0001x + 0.7774 R^2^ = 1
92	0.439 ± 0.006	y = 0.0002x + 0.4204 R^2^ = 0.9493	83	0.774 ± 0.022	y = 0.0004x + 0.7428 R^2^ = 0.7606	80	0.788 ± 0.027	y = 0.0001x + 0.7758 R^2^ = 0.9686
–	–	–	102	0.775 ± 0.019	y = 0.0003x + 0.7486 R^2^ = 0.6853	–	–	–
118	0.438 ± 0.007	y = 0.0001x +0.4241 R^2^ = 0.7745	122	0.773 ± 0.017	y = 0.0002x + 0.754 R^2^ = 0.5376	123	0.786 ± 0.024	y = 6E-05x + 0.7803 R^2^ = 0.5394
147	0.438 ± 0.024	y = 9E-05x + 0.4269 R^2^ = 0.6347	136	0.776 ± 0.015	y = 0.0002x + 0.756 R^2^ = 0.5407	–	–	–

**Table 2 molecules-25-04763-t002:** The TPC in the elaborated tinctures and herb of *S. montana.*

Active Substance	TPC, X ± SD			
	with Reference to Gallic Acid		with Reference to Rutin	
	T1	T2	T1	T2
Tincture, mg/L	2441.1 ± 78.8	1255 ± 23.0	4914.5 ± 158.74	2525.1 ± 46.21
Herb, mg/g	22.7 ± 0.73	13.0 ± 0.24	45.63 ± 1.47	26.16 ± 0.48

**Table 3 molecules-25-04763-t003:** The TFC in the tinctures (mg/L) and herb (mg/g) of *S. montana* and stability studies.

Tincture	TFC, X ± RSD		
	Immediately after Maceration	In 2 Weeks of Storage	In 6 Months of Storage
T1	943.88 mg/L ± 2.44% 8.76 mg/g	-	-
T2	613.4 mg/L ± 2.84% 6.36 mg/g	-	-
T3	614.7 mg/L ± 5.22% 6.38 mg/g	-	655.6 mg/L ± 2.41% 6.81 mg/g
T4	655.6 mg/L ± 1.36% 6.24 mg/g	-	655.6 mg/L ± 0.80% 6.24 mg/g
T5	994.1 mg/L ± 2.09% 9.47 mg/g	853.8 mg/L ± 3.19% 8.13 mg/g	973.8 mg/L ± 1.44% 9.27 mg/g
T6	508.0 mg/L ± 0.60% 5.15 mg/g	-	-
T7	523.8 mg/L ± 2.01% 5.85 mg/g	-	-

**Table 4 molecules-25-04763-t004:** Statistical analysis for comparison of the total flavonoid content (TFC) in *S. montata* herb.

No.	Comparable Samples		Comparable Mean Values of TFC, mg/g		Standard Deviations (SD) of Mean Values		${\bar{X}}_{1}-{\bar{X}}_{2}$	$S_{p}^{2}$	t	Conclusion 1	Conclusion 2
			${\bar{X}}_{1}$	${\bar{X}}_{2}$	SD_1_	SD_2_					
1	7	6	5.85	5.15	0.118	0.036	0.70	0.008	9.85	*H_0_* is rejected	The two means are statistically significantly different. There is a clear influence of the particle size on the extraction degree of flavonoids
2	7	3	5.85	6.38	0.118	0.333	0.53	0.062	2.62	*H_0_* is accepted	The two means are equal. There is no effect of the storage time of the herbal substance on the extraction degree of flavonoids
3	5	4	9.47	6.24	0.200	0.085	3.23	0.024	25.84	*H_0_* is rejected	The two means are statistically significantly different. There is a clear influence of the particle size on the extraction degree of flavonoids
4	2	4	6.36	6.24	0.18	0.085	0.08	0.020	0.70	*H_0_* is accepted	The two means are equal. There is no effect of the herb collection year on the extraction degree of flavonoids (the particle size of 3–5 mm)
5	1	5	8.76	9.47	0.213	0.200	0.71	0.043	−4.20	*H_0_* is rejected	The two means are equal. There is an effect of the herb collection year on the extraction degree of flavonoids (the particle size of 1–3 mm)
6	1	2	8.76	6.36	0.213	0.180	2.40	0.039	14.91	*H_0_* is rejected	The two means are statistically significantly different. There is a clear influence of the particle size on the extraction degree of flavonoids
7	3	3	6.38	6.81	0.333	0.164	0.43	0.069	2.01	*H_0_* is accepted	The two means are equal. There is no effect of the storage time of the tincture on the TFC
8	5	5	9.47	9.27	0.200	0.180	0.20	0.036	1.29	*H_0_* is accepted	The two means are equal. There is no effect of the storage time of the tincture on the TFC
9	5	5	9.47	8.13	0.200	0.259	1.34	0.053	7.05	*H_0_* is rejected	The two means are statistically significantly different. There is an effect of the storage time of the tincture on the TFC
10	5	5	8.13	9.27	0.259	0.180	1.14	0.050	6.33	*H_0_* is rejected	The two means are statistically significantly different. There is an effect of the storage time of the tincture on the TFC

**Table 5 molecules-25-04763-t005:** Kinetics of the extraction process for the tinctures 1 and 2.

Days of Extraction	Absorbance Mean ± SD		TFC, mg/L Mean ± SD	
	T1	T2	T1	T2
1 day	0.163 ± 0.006	0.121 ± 0.006	314.50 ± 23.02	237.08 ± 23.02
4 days	0.309 ± 0.008	0.254 ± 0.003	585.60 ± 26.73	483.45 ± 17.45
5 days	0.368 ± 0.002	0.303 ± 0.025	695.10 ± 15.60	574.42 ± 58.30
7 days	0.502 ± 0.006	0.334 ± 0.003	943.88 ± 23.02	613.41 ± 17.45

**Table 6 molecules-25-04763-t006:** Average R_f_ values of the reference standards and their respective fluorescence colours.

Reference Standard	R_f_ Value	Colour of Band			
		Before Derivatisation		After Derivatisation	
		254 nm	366 nm	254 nm	366 nm
Apigenin	0.58	Black	Black	Deep yellow	Deep yellow
Quercetin	0.56	Yellow	Yellow	Yellow	Yellow
Luteolin	0.54	Black	Black	Deep yellow	Deep yellow
Caffeic acid	0.52	Blue	Blue	Blue	Blue
Rosmarinic acid	0.34	Blue	Light blue	Blue	Light blue
(–)-Catechin	0.25	–	–	Deep black	Deep black
Chlorogenic acid	0.09	Blue	Light blue	Blue	Light blue
Rutin	0.03	Black	Black	Deep yellow	Deep yellow

**Table 7 molecules-25-04763-t007:** Chemical composition of the *Satureja montana* essential oil determined by GC-MS.

No.	Component	IR	Area, %	SD
1	1-octen-3-ol	981	1.91	0.13
2	terpinolene	1017	0.10	0.01
3	*o*-cymene	1025	1.26	0.07
4	eucalyptol	1032	0.69	0.04
5	*γ*-terpinene	1060	1.65	0.06
6	cis-*β*-terpineol	1069	1.21	0.07
7	trans-*β*-terpineol	1089	0.31	0.01
8	linalool	1099	2.09	0.21
9	thujone	1118	0.06	0.00
10	trans-*p*-menth-2-en-1-ol	1124	0.04	0.00
11	camphor	1147	0.22	0.00
12	endo-borneol	1168	0.58	0.01
13	terpinen-4-ol	1179	1.07	0.04
14	*α*-terpineol	1193	0.31	0.02
15	isothymol methyl ether	1237	0.07	0.00
16	thymol methyl ether	1246	0.27	0.01
17	thymoquinone	1256	0.46	0.05
18	linalyl acetate	1263	0.14	0.05
19	carvacrol	1289	0.05	0.00
20	thymol	1295	0.65	0.03
21	*p*-thymol	1313	81.79	0.55
22	carvacrol acetate	1375	0.23	0.01
23	caryophyllene	1422	0.89	0.02
24	*β*-cubebene	1432	0.12	0.00
25	aromadendrene	1442	0.39	0.02
26	*γ*-muurolene	1479	0.26	0.02
27	germacrene D	1484	0.61	0.02
28	ledene	1498	0.95	0.01
29	*α*-muurolene	1502	0.04	0.01
30	*β*-bisabolene	1510	0.63	0.10
31	*γ*-cadinene	1517	0.18	0.03
32	*δ*-cadinene	1526	0.35	0.02
33	spathulenol	1584	0.24	0.02
34	caryophyllene oxide	1589	0.20	0.02

**Table 8 molecules-25-04763-t008:** Characteristics of the developed tinctures of *Satureja montana* herb.

Tincture	Year of Herb Collection	Time of a Tincture Preparation	Date of a Tincture Preparation	Ratio of Herbal Substance to a Tincture	Particle Size, mm	Extraction Type
T1	2017	7 days	June 2018	1:9.3	1–3	maceration
T2	2017			1:10.4	3–5	-//-
T3	2018	30 h	August 2019	1:10.4	1–3	remaceration
T4	2018	7 days		1:9.5	3–5	maceration
T5	2018			1:9.5	1–3	-//-
T6	2018	30 h	March 2020	1:10.1	3–5	remaceration
T7	2018			1:11.2	1–3	

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
