# Peer review of "Phytochemical Evaluation of Tinctures and Essential Oil Obtained from Satureja montana Herb"

_molecules, 2020, doi:10.3390/molecules25204763_

Round 1

Reviewer 1 Report

The article Molecules-949413, titled: Phytochemical Evaluation of Tinctures and Essential Oil Obtained from Satureja Montana Herb

In this paper, the tinctures and essential oil obtained from S. montana were analyzed. The total phenolic and flavonoid contents (TPC, TFC) and phenolic compounds profile were evaluated. The authors show that the size of the sample and the extraction mode influence on the TPC and TFC. Caffeic, rosmarinic and chlorogenic acids, and rutin were identified in the tinctures using the HPTLC method. p-Thymol (81,79 %) was the predominant compound of the essential oil. The authors say the high content in polyphenols and flavonoids from S. montana tincture and essential oil could be used in the industry owed to antioxidant and antimicrobial properties. And finally, they conclude that their results can be a base for development and standardization of the tinctures and essential oil of S. montana in the industry.

In my opinion, this paper is adequate to be published with minor revision, but some issues require attention before it is acceptable. 

The authors should revise all the references.

Minor revision:

  • Line 15. Correct the spelling of Medditerrenian to Mediterranean
  • Line 38. The authors write quizine. Could it be cuisine instead of quizine?
  • Reference 5. The year is 2016
  • Line 45. The font size is different for S. montana.
  • Line 45-47. The authors say: In vitro study of the antioxidant activity of different S. montana extracts allowed to compare the most known methods [11].

Complete the phrase saying what was found comparing the methods.  

  • Line 50. The reference 5 (Tepe and Cilkiz, 2016) is not correct.
  • Line 58. Be careful with the font size.
  • In the references, some of them present the underline doi and others not
  • Table 1. Define what T1 and T2 are.
  • The authors could start explaining that they want to define what time has to be waited to obtain the mayor TFC and TPC.
  • Be careful with extra-spaces in materials and methods. For instance, lines 341 and 347.
  • Line 397-399. The numbers of lines are on the formula.
  • Line 130 Write Hajdari et al 2016, instead of (6)
  • Table 3. In T5, two weeks of storage, the units are missed. It could be supposed mg/g.
  • Line 308 write the number of the reference, (4).
  • In the conclusion. The authors should add the influence of size sample and extraction mode in TPC and TFC while the storage and collection time are not influenced in TPC and TFC.

Reviewer 2 Report

The manuscript by Hudz et al., is reporting the “Phytochemical Evaluation of Tinctures and Essential Oil Obtained from Satureja Montana Herb”. The detailed description on the experimental procedures are highly appraised. Regardless of the authors persuasive statements, the significance of the work is still not decisive. The first part seems to be relevant to the rather technological note that is important to the development of product. The second part has limited significance due to the lack of novelty, and appeared lack of scientific details compared to the first part. Though, the report will find some readers who appreciate the work.

Minors;

The reaction conditions, such as temperature and purity of solvents, should be pointed out, if the authors care about the reproducibility of TPC and TFC.

Define “FCR” when it is first shown.

Author Response

Please see the attachment. Thank you for your remarks and recommendations

Reviewer 3 Report

The ms: Phytochemical Evaluation of Tinctures and Essential Oil Obtained from Satureja montana Herb shows an interesting approach about the control analysis of the tinctures of this plant material. The different variables are well supported but there are some details that the authors should consider and in this way, the ms be more complete. Some of them are:

In your results about TPC and TFC show higher concentrations of flavonoids than the total of phenolics, how do you explain this pattern? 

In the extraction process is used as a polar solvent, and only are analyzed different aglucone forms, what about the conjugated forms of the different phenolics?

Author Response

(The authors gave the same response as above.)

Reviewer 4 Report

In the present manuscript, authors aimed to characterize the phytochemical composition of different tinctures and one essential oil obtained from Satureja Montana Herb.

The topic of the work could be of interest, but there are several issues and crucial points that need consideration. At first, the English style must be deeply improved throughout the text and some grammar, editing and typing errors have to be corrected. The content is sometimes not immediate to understand. Moreover, adding some experimental tests, even simple ones such as antioxidant, antiproliferative and antimicrobial activity could greatly increase the value and interest in the work. For that reasons, I suggest major revision of the manuscript before to assess it for publication in Molecules. In particular:

Introduction

This section should be revised making the content clearer. Biological properties should be mentioned but a long dissertation is not necessary considering that no biological assay have been carried out in present manuscript (weak point).

Materials and methods

The 3.4 Chemicals and Reagents paragraph should be the first one of Materials and Methods section. The extraction procedure should be clearly described also including a description of how the plant material was stored after harvesting. Furthermore, an explanation of how authors have measured the particle size should be added. At last, what authors call “active markers” should be defined as “standards”.

Results and Discussions

  • Table 1 is difficult to understand. Please make it clearer and easier to understand. Also, the results referred to Table 1 should be improved.
  • Is it possible to know in a clearer way the particle size of each tincture? Otherwise, results in page 5 from line 124 to line 145 are difficult to understand
  • HPLC analysis should be performed to increase the manuscript value. Indeed, total phenolic content, total flavonoid content and HPTLC give only a partial idea of the phytochemical profile. Considering that phytochemical characterization is the aim of this work it should be investigated in more depth.
  • Biological tests should be performed to increase the manuscript value

Author Response

(The authors gave the same response as above.)

Round 2

Reviewer 4 Report

Dear Editor,

I thanks the authors for the corrections made and I think that the manuscript could be accepted in present form for publication in Molecules.